# Dual polarization-enabled ultrafast bulk photovoltaic response in van der Waals heterostructures

Zhouxiaosong Zeng[1,2,3,6], Zhiqiang Tian [4,6], Yufan Wang[2,6], Cuihuan Ge[2,6], Fabian Strauß[3], Kai Braun [3], Patrick Michel[3], Lanyu Huang[2], Guixian Liu[2], Dong Li [1], Marcus Scheele [3], Mingxing Chen [4,5] ✉, Anlian Pan [1,4] ✉ & Xiao Wang [1,2] ✉

The bulk photovoltaic effect (BPVE) originating from spontaneous charge polarizations can reach high conversion efficiency exceeding the Shockley-Queisser limit. Emerging van der Waals (vdW) heterostructures provide the ideal platform for BPVE due to interfacial interactions naturally breaking the crystal symmetries of the individual constituents and thus inducing charge polarizations. Here, we show an approach to obtain ultrafast BPVE by taking advantage of dual interfacial polarizations in vdW heterostructures. While the in-plane polarization gives rise to the BPVE in the overlayer, the charge carrier transfer assisted by the out-of-plane polarization further accelerates the interlayer electronic transport and enhances the BPVE. We illustrate the concept in $MoS_2$/black phosphorus heterostructures, where the experimentally observed intrinsic BPVE response time achieves 26 ps, orders of magnitude faster than that of conventional non-centrosymmetric materials. Moreover, the heterostructure device possesses an extrinsic response time of approximately 2.2 ns and a bulk photovoltaic coefficient of 0.6 $V^{-1}$, which is among the highest values for vdW BPV devices reported so far. Our study thus points to an effective way of designing ultrafast BPVE for high-speed photodetection.

The symmetry breaking in non-centrosymmetric materials brings about intriguing physical phenomena, such as second harmonic generation[1], ferroelectricity[2], and multiple polarization states[3]. For photoelectric conversion, the symmetry-breaking-induced asymmetric distribution of photo-excited carriers can contribute an anomalous current under zero bias, which is known as the BPVE[4]. The BPVE was initially discovered and extensively studied in ferroelectric crystals due to their perceptible spontaneous polarization[5–7]. With the theoretically predicted high conversion efficiency free from the Shockley-Queisser limit[8], an open circuit voltage $V_{oc}$ far beyond the bandgap of active material[9] and a greater-than-unity quantum efficiency[10] have been demonstrated in these materials. While, in pursuit of miniaturized nanoelectronics by technological development, the increase in the effect of the depolarization field against the

[1]Key Laboratory for Micro-Nano Physics and Technology of Hunan Province, College of Materials Science and Engineering, Hunan University, Changsha 410082, China. [2]School of Physics and Electronics, Hunan University, Changsha 410082, China. [3]Institute of Physical and Theoretical Chemistry and LISA, University of Tübingen, Auf der Morgenstelle 18, 72076 Tübingen, Germany. [4]Key Laboratory for Matter Microstructure and Function of Hunan Province, Key Laboratory of Low-Dimensional Quantum Structures and Quantum Control of Ministry of Education, Synergetic Innovation Center for Quantum Effects and Applications (SICQEA), School of Physics and Electronics, Hunan Normal University, Changsha 410081, China. [5]State Key Laboratory of Powder Metallurgy, Central South University, Changsha 410083, China. [6]These authors contributed equally: Zhouxiaosong Zeng, Zhiqiang Tian, Yufan Wang, Cuihuan Ge. ✉e-mail: mxchen@hunnu.edu.cn; anlian.pan@hnu.edu.cn; xiao_wang@hnu.edu.cn

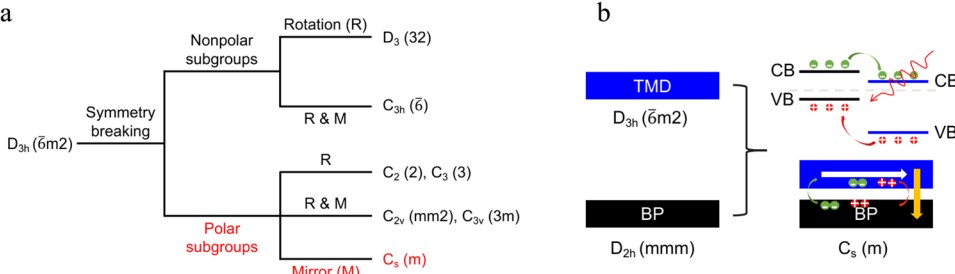

**Fig. 1 | Dual polarization in transition metal dichalcogenides/black phosphorus (TMD/BP) heterostructure. a** Symmetry breaking in TMD monolayers. Five polar groups can be obtained by breaking the $D_{3h}$ symmetry for the bulk photovoltaic effect (BPVE). **b** Polar symmetry $C_s$ obtained by stacking a TMD monolayer onto a BP monolayer. The generated in-plane and out-of-plane polarizations are illustrated by the white and the yellow arrows, respectively. The green and red arrows indicate the charge carrier transfer in conductive band (CB) and valance band (VB), respectively.

spontaneous polarization suppresses the BPVE in conventional ferroelectric crystals at a thin limit and hinders their further application.

In addition to conventional ferroelectric crystals, recently, preserved spontaneous polarizations have been observed in emerging vdW semiconductor materials[11–13]. With the spatial designability and constructability, vdW materials through manipulating stacking orders[14] and angles[15] or applying a strain[16,17] can break their centrosymmetry and hence exhibit advances of the BPVE. Benefiting from their decreased thickness comparable to the free path length, the polar two-dimensional (2D) vdW materials and structures have shown greatly enhanced BPV current density and coefficient[18]. While due to the directional carrier transporting along the nano-size tube or the quantum well, one-dimensional vdW structures[19–21] are reported to have high-performance BPVE. Besides, combining the different vdW materials with complementary properties, such as conductive types, to form heterostructures and engineering the low-symmetric interfaces that facilitate the charge extraction may enable a further progress of the BPVE and nano devices for a specific application. However, understanding the carrier dynamics of vdW materials in BPVE generation at the low-dimensional limit and realizing a BPV device with fast speed performance remains a challenge.

Here, we propose a symmetry-engineered structure of MoS₂/black phosphorus (BP) with in-plane and out-of-plane polarizations that combines different photocurrent generation mechanisms. A spontaneous current along the polar direction with a high ratio of anisotropy is revealed by the scanning photocurrent imaging with sub-MoS₂-bandgap illumination. By combining charge carrier distribution simulation and time-resolved photocurrent (TRPC) measurements, we find that the heterostructure exhibits an ultrafast response time of 26 ps for the intrinsic BPVE generation, which results from the synergistic effect of the in-plane polarization and carrier accumulation assisted by the vertical built-in electric field. In comparison, the WSe₂/BP BPV devices without the accumulation possess an intrinsic response time of approximately 1 ns, which is fifty times slower than that for the MoS₂/BP devices. The synergistic effect in our MoS₂/BP heterostructure further induces a 2.2 ns extrinsic response time and one of the highest BPVE coefficients (up to 0.6 V⁻¹) among the existing vdW BPV devices. The large BPV coefficient, fast response speed, and wide detection range make our MoS₂/BP BPV device promising for the next generation of the self-powered ultrathin photodetector with high efficiency.

## Results
### Design of MoS₂/BP heterostructure with dual polarization
The observation of BPVE relies on the broken inversion symmetry, for which polar groups can be introduced. Figure 1a shows that breaking the $D_{3h}$ symmetry of the transition metal dichalcogenides (TMD) monolayers can give rise to five types of polar groups, among them $C_2$, $C_{2v}$, and $C_s$ can induce in-plane polarization. Besides strain

engineering[22] and effective dimensionality reduction[23], such broken symmetries can be easily achieved by building heterostructures. For instance, the BP monolayer has a distinctly different lattice and symmetry ($D_{2h}$) from the TMD monolayers. As a result, interfacing a TMD monolayer with a BP monolayer to form a TMD/BP heterostructure can lead to a symmetry reduction for the TMD monolayer from the $D_{3h}$ symmetry to $C_s$ with merely the mirror symmetry left. On the other hand, the performance of the BPVE is associated with the extraction of the charge carriers. In this regard, the built-in electric field together with the band alignment of the two constituents plays a key role in the transport and kinetics of photogenerated carriers. The ideal situation is that the built-in electric field separates the photogenerated carriers in both the overlayer and the substrate, and yields an accumulation of the carriers in each constituent. For TMD/BP heterostructure, the combination of the type-II band alignment and the build-in electric field pointing from the TMD layer to BP would favor this process (Fig. 1b). Based on the above theoretic analyses, we propose a MoS₂/BP heterostructure with dual interfacial polarizations (other TMD/BP heterostructures see Supplementary Note 1).

We prepared MoS₂/BP heterostructures by elaborately stacking the MoS₂ monolayer on the BP flake with aligned armchair directions (Fig. 2a), where the crystal directions were determined (Supplementary Note 2) according to the second harmonic generation (SHG) polarization in MoS₂ monolayer (Fig. 2b left panel) and the Raman intensity ratio in BP (Fig. 2b, right panel). The BP flake and MoS₂ monolayer in respective p-type and n-type conductive behaviors were confirmed by the transfer characteristic measurements (Supplementary Fig. 4). The planar averaged electrostatic potential simulation along the z direction (Fig. 2c) and Kelvin probe force microscope (KPFM) measurement (Fig. 2d) simultaneously determined the out-of-plane polarization strength at tens of microvolt.

We fabricated a MoS₂/BP device with the electrodes parallel to the mirror plane of the heterostructure and investigated its photoresponse. As a comparison, an armchair direction aligned WSe₂ monolayer/BP heterostructure device with the same configuration was also prepared, where the out-of-plane charge carrier extraction is unfavored (Supplementary Note 3)[15]. Under the illumination of a 633 nm continuous wave (CW) laser, the MoS₂/BP device shows a short-circuit current of approximately 250 nA, doubling that of the WSe₂/BP device (Fig. 2e). With the increase in the input laser power, both devices display a linear to sub-linear (0.5) photocurrent transition at an excitation power of approximately 50 μW (Fig. 2f), which is different from the photovoltaic (PV) effect trend originating from the p-n junction or Schottky barrier[19]. We further used a linearly polarized laser to check the polarization of the generated photocurrent. With the rotation of the input laser, the photocurrent response in the MoS₂/BP device displays a distinct anisotropy with its maximum along the mirror plane direction, demonstrating the

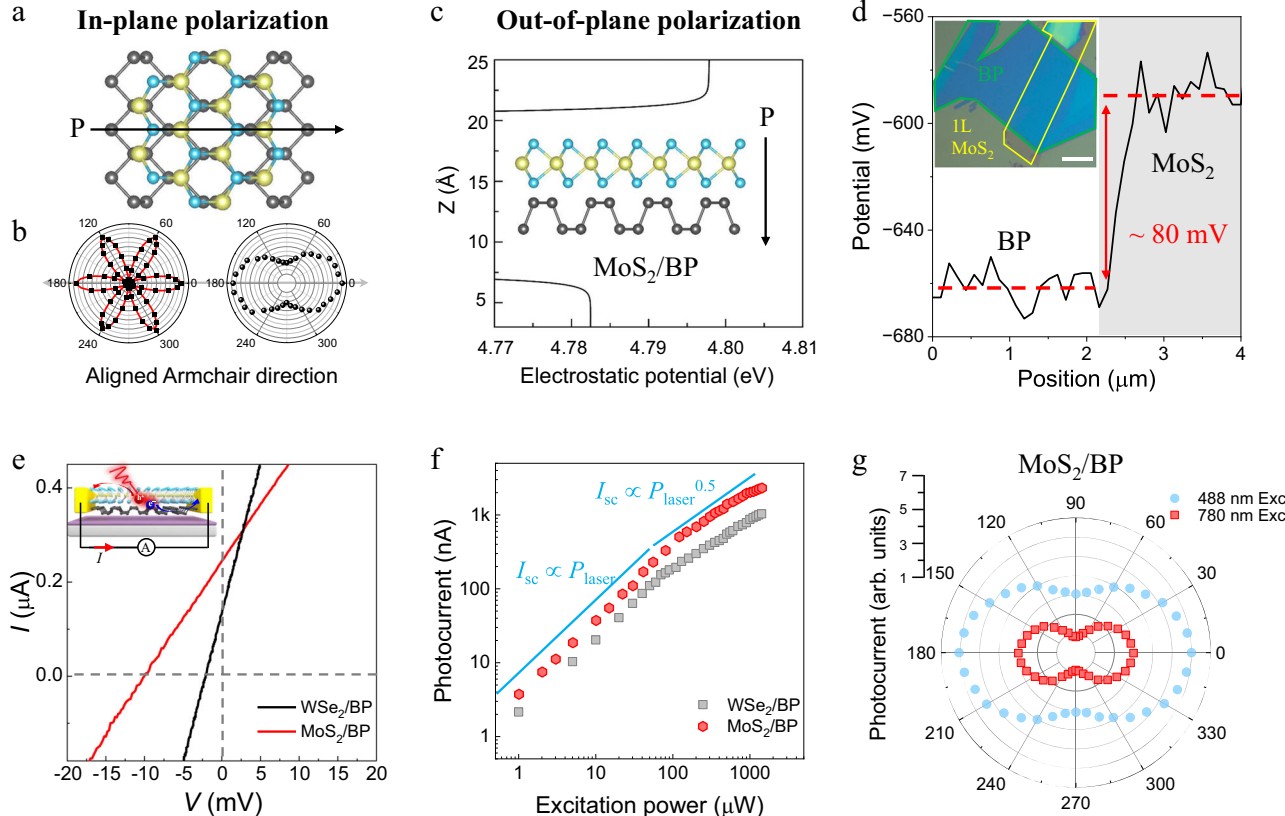

**Fig. 2 | Design and BPVE of MoS$_2$/BP heterostructure with in-plane and out-of-plane polarizations. a** Schematic illustration of the in-plane polarization generation (indicated by the black arrow) in the armchair direction aligned MoS$_2$/BP heterostructure. **b** Second harmonic generation (SHG) polarization in monolayer MoS$_2$ (left panel) and Raman intensity ratio of the BP peak $A_g^2/A_g^1$(right panel), where the polar direction indicated by the gray arrow implies the determination of the armchair direction in two different materials respectively. **c** Planar averaged electrostatic potential simulation along the z direction in the armchair direction aligned MoS$_2$/BP heterostructure implying a generation of the out-of-plane polarization (indicated by the black arrow). The inset shows the schematical side view of monolayer MoS$_2$/BP heterostructure. **d** Kelvin probe force microscope (KPFM) line scan profile of MoS$_2$/BP heterostructure showing the relative surface potential difference. The inset shows the optical image of the measured sample with an aligned armchair direction, where the green and yellow solid lines highlight the position of MoS$_2$ monolayer and BP. The scale bar is 8 μm. **e** Comparison of the photocurrent in WSe$_2$/BP (black) and MoS$_2$/BP (red) devices under the 633 nm continuous wave (CW) laser illumination with power density $P$ = 71.3 mW/cm$^2$. The inset shows the schematic illustration of the TMD/BP BPV device. **f** Input laser power dependence of photocurrent intensity in MoS$_2$/BP (red dots) and WSe$_2$/BP (gray squares) devices. The solid lines serve as guidelines for linear and square-root dependence. **g** Photocurrent anisotropy in MoS$_2$/BP device with the illumination of 488 nm CW laser (light blue) and 780 nm pulse laser (red).

broken rotation symmetry of the integrated heterostructure. Interestingly, a photoresponse was obtained under the 780 nm illumination (sub-MoS$_2$ monolayer bandgap excitation) and the ratio of anisotropy from it was larger than that from 488 nm excitation (above-MoS$_2$ monolayer bandgap excitation) (Fig. 2g).

## BPVE generation with sub-MoS$_2$ monolayer bandgap illumination

To determine whether the photoresponse under 780 nm (1.589 eV) illumination is derived from the BPVE, scanning photocurrent microscope (SPCM) measurements were performed without external bias. For a better demonstration of the photocurrent generation, we fabricated another MoS$_2$/BP device with two pairs of crossed electrodes that were parallel (E1-E2) and perpendicular (E3-E4) to the mirror plane, respectively (Fig. 3a). The atomic force microscope (AFM) line profiles indicate that the thickness of BP was 15 nm and the thickness of MoS$_2$ was 0.7 nm (Fig. 3d) for the fabricated device. With connected E1-E2 electrodes, photocurrent with unchanged polarity appeared at the entire heterostructure region and the photocurrent intensity became higher far away from the electrodes (Fig. 3b), reflecting a spontaneous charge carrier separation by the in-plane polarization. In contrast, with the connected E3-E4 electrodes, a weak photocurrent

with different polarities was obtained at the near electrode regions (Fig. 3c), which was attributed to the photothermoelectric effect[24]. These phenomena could be more distinctly observed in the extracted photocurrent line profiles (Fig. 3e, f), where the photocurrent from electrodes E1-E2 was one order of magnitude larger than that of electrodes E3-E4. The reproducibility of these phenomena is provided in Supplementary Note 6. The comparison of the photocurrent pattern and intensity in different directions reflects a clear BPVE under a sub-monolayer MoS$_2$ bandgap (1.82 eV) excitation[25].

We conducted density-functional theory (DFT) calculations to analyze the generation of in-plane polarization related to this BPVE. The in-plane polarization can be illustrated by the polar charge density ($\Delta\rho$), which is shown in Fig. 3g. For the freestanding MoS$_2$, one can see that $\Delta\rho$ shows the three-fold rotation symmetry (Supplementary Fig. 9a), which is, however, absent for the MoS$_2$/BP heterostructure. This asymmetry results in a net in-plane polarization, which can also be seen from the planar-averaged $\Delta\rho$ (Fig. 3h). One can see that there is a separation of the positive and negative charge centers. In addition, we further show that $\Delta\rho$ is mainly distributed at the overlayer, i.e., MoS$_2$ (Fig. 3i). We calculated the in-plane polarization using the Berry phase method, which gives 0.14 pC/m for the MoS$_2$/BP heterostructure (Supplementary Note 9). This value is comparable to that for bilayer γ-InSe[26].

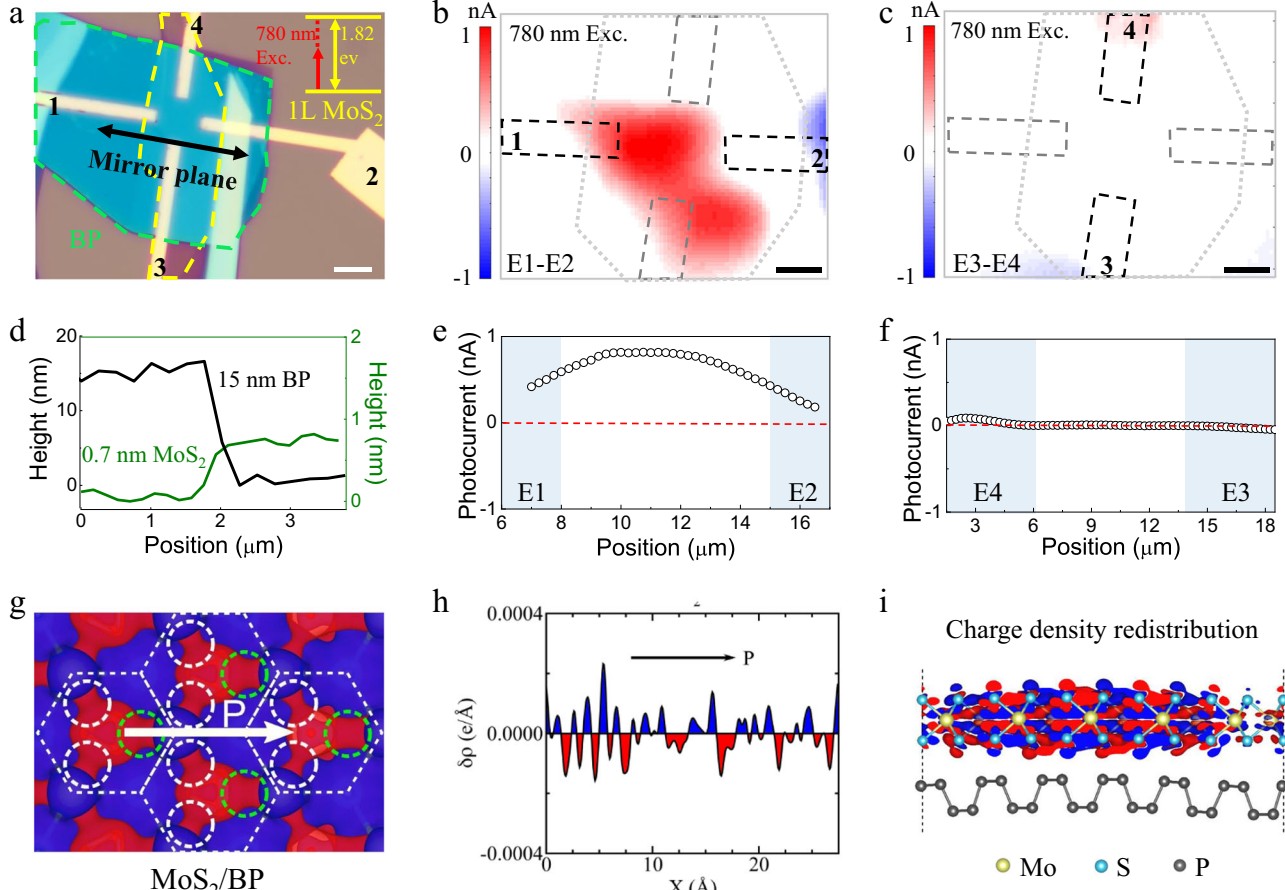

**Fig. 3 | BPVE generation under sub-MoS₂-bandgap illumination. a** Optical image of the MoS₂/BP BPV device with two pairs of electrodes, where the electrodes E1-E2 are parallel to the mirror plane of the device, while the E3-E4 are perpendicular to the mirror plane. The yellow and green curves outline the monolayer MoS₂ and BP, respectively. The inset shows that the excitation with 780 nm (1.589 eV) is below MoS₂ bandgap (1.82 eV). The white scale bar is 7 μm. **b, c** Scanning photocurrent microscope (SPCM) images with the electrodes E1-E2 (**b**) and E3-E4 (**c**) with the 780 nm laser. The dashed gray lines highlight the overlap region of the heterostructure. The black scale bars are 4 μm. **d** AFM line profiles of the MoS₂/BP device indicating the thicknesses of MoS₂ and BP being 0.7 nm and 15 nm, respectively.

**e, f** Photocurrent line profiles along the electrodes E1-E2 (**e**) and E3-E4 (**f**) at the MoS₂/BP heterostructure region. The shaded areas indicate the positions of electrodes. **g** Calculated polar charge density with electron accumulation (blue) and depletion (red) in MoS₂/BP heterostructure. The dashed hexagon represents the unit cell of MoS₂. The dashed circles show the symmetry breaking at the heterostructure interface. The white arrow denotes the charge polarization. **h** Planar averaged differential charge density in MoS₂/BP heterostructure. **i** Side view of the in-plane polarization distribution simulation in MoS₂/BP heterostructure, where the charge redistribution mainly occurs in MoS₂ layer.

## Ultrafast intrinsic response and dynamic analyses of BPVE

To further elucidate the origin of the BPVE, we performed detailed intrinsic current dynamic measurements in the MoS₂/BP heterostructure with the time-resolved photocurrent (TRPC) technique (Fig. 4a). For a better understanding and comparison of the photocurrent generation mechanisms, WSe₂/BP BPV devices and pure BP two-terminal PV devices with different BP thicknesses were measured correspondingly. The spontaneous photocurrent derived from BPVE in MoS₂/BP and WSe₂/BP devices was collected at the center of the heterostructures while the spontaneous photocurrent derived from Schottky barrier in pure BP two-terminal devices was collected at the edge of electrodes (red dots in Fig. 4c). All intrinsic response time measurements were conducted at zero external bias. For BP two terminal devices, the intrinsic response time extended linearly from 68 ps to 488 ps with the increase in BP thickness from 4 nm to 40 nm (middle panel in Fig. 4b and black dots in Fig. 4d). We attribute this phenomenon to a defect-related recombination process[27,28], where the photo-generated charge carriers in the middle of the thicker BP are captured from the defects with a longer recombination center than that of the surface. Meanwhile, these carriers would experience a longer out-of-plane distance to drift to

the electrodes on the top. In contrast to the prominently extended response time in pure BP, the BPVE (Supplementary Note 10) dynamics from WSe₂/BP heterostructures demonstrated intrinsic response times of approximately 1 ns (upper panel in Fig. 4b and orange dots in Fig. 4d), which are not sensitive to the thickness of BP. This phenomenon can be understood in that the generation of the photocurrent is due to a broken-symmetry-induced interfacial behavior as the in-plane polarization only distributes within a few atomic layers from the heterointerface[15]. After excitation, the charge carriers from the interface are separated by the in-plane polarization and form a BPVE. The slightly increased response time for over 40 nm WSe₂/BP heterostructures could be attributed to the influence of the increased amount of generated charge carriers in the bottom BP, which may affect the dynamics of carriers originating from the BPVE. The two distinct photocurrent dynamic results in WSe₂/BP and BP in turn demonstrate the different origins between the BPVE and the PV effect in respective structures. Intriguingly, regarding MoS₂/BP devices, we observed the fastest 26 ps intrinsic response time (lower panel in Fig. 4b), much shorter than that in WSe₂/BP heterostructures. In addition, these intrinsic response times display a linear dependence on the BP thickness (light blue

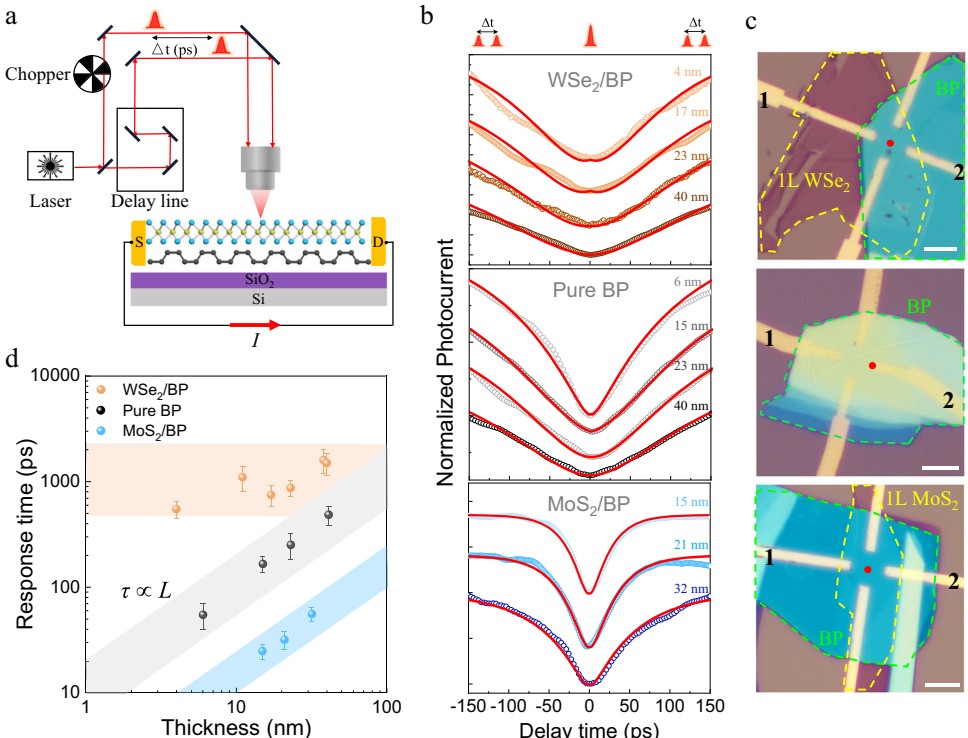

**Fig. 4 | Intrinsic photocurrent response in BPV devices via time-resolved photocurrent (TRPC) technique. a** Schematic illustration of the TRPC measurement. The $\Delta t$ indicates the delay time between pump and probe beams. **b** TRPC measurements of WSe$_2$/BP BPV devices, pure BP two-terminal PV devices, and MoS$_2$/BP BPV devices with different BP thicknesses. The hollow circles are experimental data and the solid red lines show the fitting of the data. **c** Representative optical images in different structures showing the signal collection positions at red dots. The green and yellow dash lines indicate the position of BP and monolayer TMDs. All scale bars are 6 μm. **d** Comparison of the BP thickness ($L$) dependence of the intrinsic response time ($\tau$) in WSe$_2$/BP BPV devices (orange dots), pure BP two-terminal PV devices (black dots), and MoS$_2$/BP BPV devices (light blue dots). The orange, gray and light blue areas serve as guides for different BP thickness dependence of the intrinsic response time. Error bars correspond to the standard deviation from multiple measurements.

dots in Fig. 4d), similar to that of the PV effect in pure BP devices. These observed phenomena in MoS$_2$/BP are consistent with our proposed carrier extraction design (Fig. 1b). With the type-II band alignment and vertical p-n junction in the heterostructures, the electrons generated from the BP layer by the illumination are rapidly transferred into the monolayer MoS$_2$ under the out-of-plane polarization. After that, the formed in-plane polarization can further accelerate these electrons via the BPVE. Because the electrodes parallel to the mirror plane also contact the bottom BP flake in our structures, the remaining holes in the bottom BP are collected by the electrodes and the carrier circulation is realized. In this way, the BPVE response for MoS$_2$/BP heterostructure is related to the BP thickness, and its ultrafast speed is attributed to the charge carrier transfer.

Regarding the mechanism of spontaneous photocurrent, shift current is one of the main originations for BPVE. The generation of shift current in bulk non-centrosymmetric materials and TMD has been investigated by the calculation[29] and THz emission spectroscopy[30–32], where its typical response time was roughly 0.1 ps. However, in the shift current mechanism, displacement of electrons occurs during the optical transition process, and it does not have a contribution to the BPVE after the optical transition. Besides electron shift in real space, the BPVE is also related to the unbalanced velocity distribution in momentum space, which is referred to ballistic current. Normally, under linearly polarized light excitation, the ballistic current can occur in non-centrosymmetric non-magnetic materials, if an additional scattering process is considered[33]. For our MoS$_2$/BP heterostructures, the BPVE is not only contributed by the photo-excited charge carriers at the hetero-interface but also related to the charge carriers transferred from the bottom BP, where the charge carrier transfer process

may introduce electron-phonon or electron-defect interactions and contribute ballistic current.

## High-frequency extrinsic photoresponse

In contrast to the intrinsic response measurement which provides the upper limit speed of the device's photoresponse, the extrinsic response investigation gives the conventional device speed and reflects the ability of a photodetector to process high-frequency signals. To this end, we determined the extrinsic response time in our MoS$_2$/BP device using the experimental setup illustrated in Fig. 5a (details see Methods and Supplementary Notes 14 and 15). Upon the 636 nm pulse illumination, the device displays a constant response for several cycles as shown in Fig. 5b. We exemplarily extract a single pulse response and resolve it on a semi-log scale axis. The 90% to 10% photocurrent decay indicates an extrinsic response time of 2.2 ns (Fig. 5c, black curve). Fast Fourier transformation (FFT) was used to calculate the electrical bandwidth under the single excitation frequency and a 3 dB bandwidth of approximately 150 MHz was obtained (Fig. 5d, gray squares), which is comparable to that of state-of-art self-powered photodetectors[34,35]. We further changed the input laser to 779 nm and conducted laser repetition frequency dependent measurements. The results show a comparable response time (Fig. 5c, red curve) and 3 dB bandwidth (Fig. 5d, red dots) for all laser frequencies (Fig. 5e, f). These results further illustrate that the unique BPVE generation in our MoS$_2$/BP heterostructure device can be used to detect near-infrared signals.

## Overview of BPV coefficient and response time

We now quantitatively compare the BPVE performance of our MoS$_2$/BP heterostructure with other recently reported vdW materials and

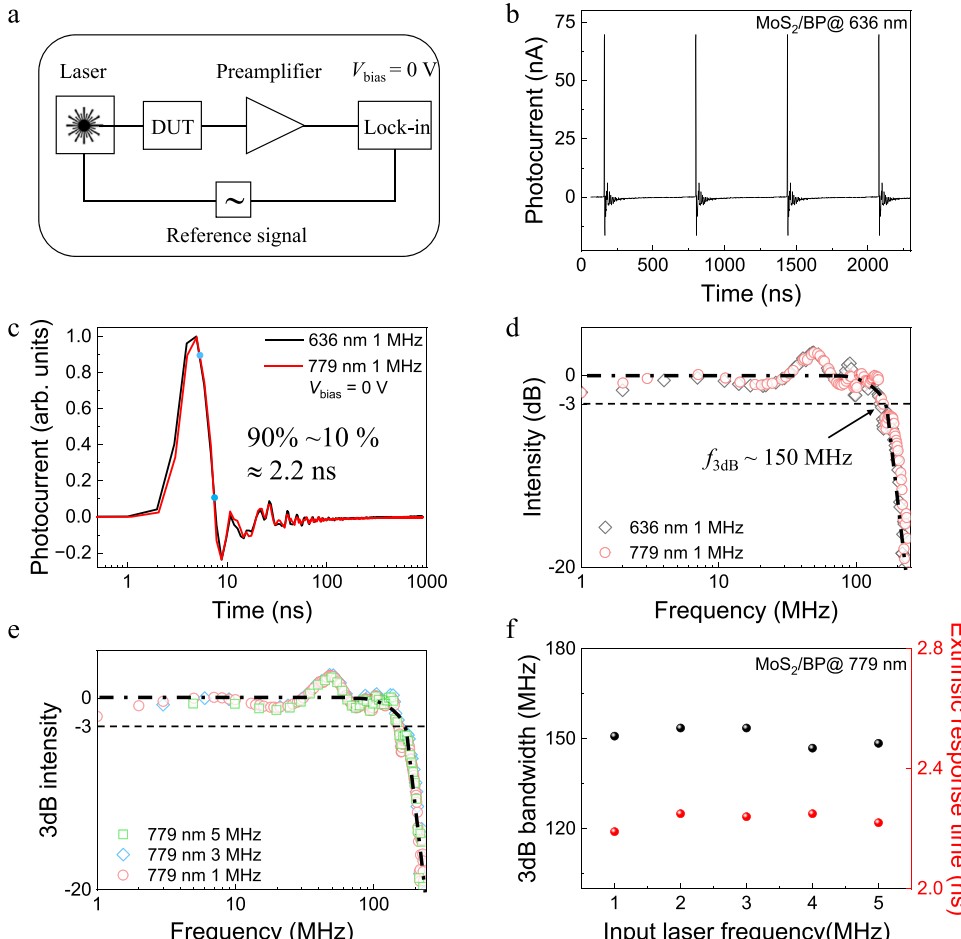

**Fig. 5 | Extrinsic photocurrent response in MoS₂/BP BPV device. a** Experimental configuration for the extrinsic response time measurements. DUT device under test. **b** Photoresponse towards 636 nm pulse sequence under zero external bias. **c** Normalized extrinsic response time results in semi-log scale axis towards 636 nm (black) and 779 nm (red) impulse laser. The extrinsic response time defined by the 90% to 10% photocurrent decay (indicated blue dots) are approximately 2.2 ns for two different lasers. **d** Fourier transformed 3 dB bandwidth corresponding to the results in (**c**). **e** Fourier transformed 3 dB bandwidth results towards 779 nm impulse laser for different repetition frequencies. **f** Input 779 nm laser frequency dependence of extrinsic response time and the 3 dB bandwidth.

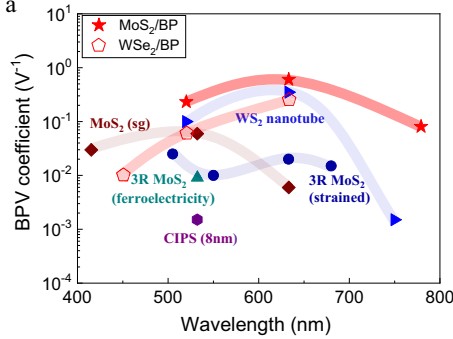

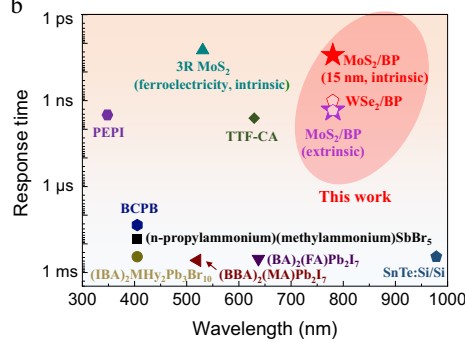

**Fig. 6 | BPV coefficients and response time in various symmetry breaking devices. a** Comparison of the BPV coefficient in various vdW BPV devices. Data for other materials are taken from the literature (WS₂ nanotube, ref. 19; MoS₂ (strain gradient), ref. 3, 16. R MoS₂ (ferroelectricity), ref. 3, 14. R MoS₂ (strained), ref. 17; CuInP₂S₆ (CIPS) (8 nm), ref. 18). The calculated maximum BPV coefficient of our MoS₂/BP heterostructure is approximately 0.6 V⁻¹, which is among one of the highest values in various vdW BPV devices. **b** Comparison of the response time between our TMD/BP heterostructures and other conventional ferroelectric materials. Data for other materials are taken from the literature. (3R MoS₂ (ferroelectricity), ref. 36; (iso-pentylammonium)₂(ethylammonium)₂Pb₃I₁₀ (PEPI), ref. 40; tetrathiafulvalene-*p*-chloranil (TTF-CA), ref. 41; (C₆H₅CH₂NH₃)₂CsPb₂Br₇ (BCPB), ref. 42; (n-propylammonium)(methylammonium)SbBr₅ ref. 43; (IBA)₂MHy₂Pb₃Br₁₀, ref. 44; (BA)₂(FA)Pb₂I₇, ref. 45; SnTe:Si/Si, ref. 46; (BBA)₂(MA) Pb₂I₇, ref. 47).

ferroelectric materials. In an evaluation of the BPVE strength in our MoS₂/BP heterostructure, we calculated the BPV coefficient based on: $j_i = \beta_{ilm} E_l E_m^* I$, where $j_i$ is the current density for BPVE, $\beta_{ilm}$ is the BPV coefficient, $E_l$ and $E_m^*$ are the light polarization unit vectors and $I$ is the

light intensity. Here we used the 633 nm, 520 nm, and 779 nm CW laser with the excitation power of 71.3 mW/cm². For the MoS₂/BP device with a channel length of 7 μm and a photosensitive area of 49 μm, the calculated BPV coefficients were 0.6 V⁻¹, 0.23 V⁻¹ and 0.08 V⁻¹,

respectively. Comparing the BPV coefficient with that reported in other vdW materials, for instance, 3R-MoS$_2$, WS$_2$ nanotube, and CuInP$_2$S$_6$ (CIPS) (Fig. 6a), we found that our MoS$_2$/BP device demonstrates one of the highest values, indicating the improved strength by making use of dual interfacial polarizations. Besides, the short-circuit current density $j_i$ is also plotted as the function of incident power density $P$ (Supplementary Note 17), where our MoS$_2$/BP heterostructure is also among the highest results.

In terms of speed performance, the extrinsic response time of our MoS$_2$/BP (15 nm BP) device is around 2.2 ns, which is five orders of magnitude faster than that of most recently reported ferroelectric perovskite at a time scale of approximately ten to hundreds of microseconds (Fig. 6b). In addition, the intrinsic response time down to 26 ps is comparable to that in graphene-incorporated 3R MoS$_2$[36]. This intrinsic response time corresponds to a photodetection bandwidth of $f = 0.55/\tau = 21$ GHz, which almost reaches the highest reported photo-switching speed for BPV photodetectors. It should be noted that further improved response speed can be reached with decreasing the BP thickness.

In summary, we have demonstrated a high efficiency and ultrafast BPVE in polarization-engineered MoS$_2$/BP van der Waals heterostructure device. With the synergic effect of in-plane and out-of-plane dual-polarization, the heterostructure displays a 0.6 V$^{-1}$ BPV coefficient and an intrinsic 26 ps response time, which suggests a possibility for the enhancement of the photoresponse by combining different photocurrent generation mechanisms. Moreover, a 2.2 ns extrinsic response time and 150 MHz 3 dB bandwidth independent of the laser frequencies are obtained, demonstrating the potential of the vdW BPV device for high-frequency photoswitching applications. Our results provide a perspective of the BPV device for the ultrathin self-powered photodetectors with fast response speed.

## Methods
### Device fabrication
The pure BP devices and TMD/BP devices were both fabricated based on mechanically exfoliated materials with the all-dry transfer method. Before transfer, BP or TMD flakes were first mechanically exfoliated from single crystals (purchased from Nanjing Muke Nanotechnology Co., Ltd.) onto transparent polydimethylsiloxane (PDMS), and the crystal directions of BP flakes and TMD flakes on PDMS were determined via polarized Raman spectrum and polarized SHG, respectively. Thereafter, the BP flake was placed onto a silicon substrate with a 300 nm-thick silicon dioxide layer. The TMD flake was then aligned to the BP flake and transferred onto it with the help of a microscope. Cr/Au (10 nm/50 nm) conducting electrodes on top of 2D BP or TMD/BP with 7 μm channel length were fabricated using standard electron beam lithography (EBL), metal thermal evaporation, and lift-off processes.

### Basic characterizations
Atomic force microscopy (AFM) (Bruker Dimension Icon) in the tapping mode was used to identify the thickness of the samples. Raman measurements of the samples were taken using a confocal microscope (WITec, alpha-300) equipped with a 50× objective lens (Zeiss EC Epiplan). The excitation source of the Raman was a 488 nm continuous-wave laser, and the laser beam was focused to the size of about 1 μm on the samples. The SHG measurements of the samples in reflection configuration were performed using the same confocal microscope with 800 nm laser pulses (repetition rate of 80 MHz, pulse width of 100 fs). The electrical properties were measured with an Agilent-B1500 semiconductor analyzer in a LakeShore vacuum chamber of 10$^{-4}$ Pa. The extrinsic response measurements were performed on a probe station using a picosecond pulse laser drive (Taiko PDL M1, PicoQuant) equipped 636 nm and 779 nm laser head with a pulse width of < 500 ps. The generated photocurrent was pre-amplified with a FEMTO HSA-Y-1-60 high-speed current amplifier,

after that, it was collected with a Zurich Instruments UHFLI lock-in amplifier.

### SPCM and TRPC measurements
SPCM and TRPC were performed on our home-built setup. In SPCM measurements, a 780 nm fiber laser (NPI Rainbow 780 OEM) with a pulse width of 80 fs or a 488 nm continuous wave laser was chopped by a mechanical chopper at 1050 Hz, and then focused onto the sample by a long working distance objective (Olympus LMPLFLN 50×) near the diffraction limit. The generated photocurrent was collected by a lock-in amplifier (Stanford SR830) at the chopped frequency with a background noise of approximately 0.2 pA. The SPCM measurements with the resolution close to the diffraction limit were performed by raster scanning the entire device mounted on a piezoelectric translation stage (Piezoconcept LT3) according to the fixed laser spot. In TRPC studies, a 780 nm pulse laser was split into two beams to form a pump-probe measurement configuration, and the probe beam was chopped so that the lock-in amplifier could only measure its photocurrent. The pump beam was delayed by different path lengths, with the delay time precisely controlled by a mechanical delay stage (Thorlabs DDSM100/M). The pump and probe beams were recombined after the delay line stage, and focused onto the sample using the same objective. The temporal resolution of the TRPC set-up is approximately 1 ps. The electric field of the input laser was always parallel to the armchair direction of the BP flakes or heterostructures in all TRPC measurements to ensure photocurrent saturation.

### Intrinsic response time fitting
Global fitting of the TRPC signals was performed using the equation: $\frac{PC(\Delta t)}{PC(\Delta t \to \infty)} = 1 - Ae^{\frac{-|\Delta t|}{\tau}}$, where amplitudes A and time constants τ are the fitting parameters. The exponential time constant $\tau$ gives the intrinsic response time of the devices.

### DFT simulation
A slab structure was used to model the BP monolayer, TMD monolayers, and their heterostructures. A vacuum distance of 20 Å was set between adjacent slabs to avoid interlayer interaction in the periodic boundary condition. The in-plane lattice constants of the BP monolayer are 4.626 Å for $a$ and 3.299 Å for $b$, respectively. As for WSe$_2$ and MoS$_2$ monolayers, the lattice constants are 3.323 Å and 3.184 Å, respectively. In the heterostructures, a $5 \times 1$ supercell was used for the BP monolayer, which has a small lattice mismatch (<1%) with a $4\sqrt{3} \times 1$ supercell of WSe$_2$. For the heterostructures of BP/MoS$_2$, the BP is in a $6 \times 1$ supercell and the MoS$_2$ monolayer has a $5\sqrt{3} \times 1$ supercell. Density-functional theory (DFT) calculations were performed using the Vienna ab initio Simulation Package (VASP)[37]. The projector augmented wave method was used to construct pseudopotentials[38]. The plane-wave energy cutoff was set to 400 eV. The exchange-correlation functional was treated using the generalized gradient approximation as parametrized by Perdew, Burke, and Ernzerhof[39]. The Brillouin zone was sampled by $12 \times 12 \times 1$ and $2 \times 12 \times 1$ k-meshes for freestanding TMD monolayers and the heterostructures, respectively. Structural relaxations were done with a threshold of 10$^{-2}$ eV Å$^{-1}$ for the residual force on each atom. In the calculations of heterostructure, vdW dispersion forces between the adsorbate and the substrate were accounted for by using the semi-empirical DFT-D3 method. The polarizations were computed by using the Berry phase method. To investigate the interface effects on the charge redistribution in TMD monolayers, we define the polar charge density as $\Delta\rho = \Delta\rho_i - \Delta\rho_f$, where $\Delta\rho_i$ and $\Delta\rho_f$ represent the density difference for the interfaced and freestanding TMD monolayers, respectively. $\Delta\rho_i$ is defined as $\Delta\rho_i = \rho_{i\_tot} - \rho_{i\_TM} - \rho_{i\_X} - \rho_{BP}$, where $\rho_{i\_tot}$ represents the total charge density of the TMD/BP heterostructures, $\rho_{i\_TM}$, $\rho_{i\_X}$, and $\rho_{BP}$ denote the ones of the isolated Mo/W, S/Se, and BP monolayer in the interface structures, respectively. $\Delta\rho_f$ is calculated

by $\Delta\rho_f = \rho_{f\_tot} - \rho_{f\_TM} - \rho_{f\_X}$, where $\rho_{f\_tot}$, $\rho_{f\_TM}$, and $\rho_{f\_X}$ denote the charge densities of the freestanding TMD monolayer, isolated Mo/W and S/Se atoms in the system, respectively.

## Data availability

Relevant data supporting the key findings of this study are available within the article and the Supplementary Information file. All raw data generated during the current study are available from the corresponding authors upon request.

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

## Acknowledgements

This work was financially supported by the National Key Research and Development Program of Ministry of Science and Technology (Nos. 2022YFA1204300 to A.P.), the National Natural Science Foundation of China (Nos. U23A20570, 92263107, and 52022029 to X.W.; 52302175 to Z.Z.; 52221001 and 62090035 to A.P.; 12174098 to M.C.), the Hunan Provincial Natural Science Foundation of China (Nos. 2023JJ40138 to Z.Z.; 2022JJ30142 to X.W.), the China Postdoctoral Science Foundation (2022M721081 to Z.Z.), the Alexander von Humboldt foundation with a postdoctoral fellowship to Z.Z., the Deutsche Forschungsgemeinschaft (DFG) under grant SCHE1905/9-1 (project no. 426008387 to M.S.) as well as the by the European Research Council (ERC) under the European Union's Horizon 2020 research and innovation program (grant agreement No 802822 to M.S.). Calculations were carried out in part using computing resources at the High Performance Computing Platform of Hunan Normal University.

## Author contributions

X.W. conceived the original project. Z.Z. and K.B. built the experimental set-up. Z.Z., C.G., L.H., and G.L. prepared the materials. Z.Z. and Y.W. fabricated the devices with the input from D.L., Z.Z., F.S., and P.M. carried out the extrinsic photoresponse measurements. Z.Z. carried out the TRPC and SPCM measurements and performed data analysis. Z.T. and M.C. carried out the simulations. Z.Z., X.W. and M.C. wrote the manuscript. X.W., A.P., M.C., and M.S. supervise the work. All authors contributed to the general discussion.

## Competing interests

The authors declare no competing interests.
