## [Peer Review File · Nature Communications]

Dual polarization-enabled ultrafast bulk photovoltaic response in van der Waals heterostructuresREVIEWER COMMENTS

Reviewer #1 (Remarks to the Author):

Bulk photovoltaic effect in vdW materials attracts great attention in recent years. In this work, authors built 1L-TMDCs/BP heterostructures, reproduced previous observation of BPVE in 1L-TMDCs/BP, and demonstrated ultrafast bulk photovoltaic response. Although, the BPVE in 1L-TMDCs/BP heterostructures has been demonstrated in previous works by Iwasa et al., authors provided some interesting results from a new perspective: the ultrafast response speed due to both existence of in-plane and out-plane polarization in the MoS₂/BP heterostructure. Major revisions should be made and data analysis should be strengthened before it can be considered for publications. Here is my comments.

1. Authors claim that intrinsic response time of WSe₂/BP structure is independent of the thickness of BP. However, from Fig. 3c, it is obvious that the response time also increases with increasing thickness of BP.

2. Did author measure the KPFM line scan profile of WSe₂/BP? Is there any potential difference?

3. In abstract, authors claim that “the quantum tunneling assisted by the ...”. How does quantum tunneling phenomenon occur in the out-of-plane direction? I cannot see any quantum tunneling effect in this device configuration.

4. For characterizing the response time of BP, MoS₂/BP, and WSe₂/BP devices, authors should provide clear characterization condition, such as bias voltage. If there is, will bias voltage affect the response time?

5. What happens for response speed using above-bandgap lasers?

6. In method section, authors claim that the heterostructures were built after Raman characterization of BP flakes. Since surfaces of BP flakes are immediately oxidized when touching the air., how can authors guarantee a clear and oxidation-free interface which is very crucial for BPVE observation.

7. Can author explain why the value of BPVE photocurrent is maximum in the middle region of the channel?

8. How many layers of BP for DFT calculations? Will it affect the calculation results and conclusions?

9. Authors may include and discuss two recent works about BPVE in vdW materials and heterostructures Nat Commun 14, 4230, 2023 and Nat Commun 15, 501, 2024.

Reviewer #2 (Remarks to the Author):

The paper “Dual polarization enabled ultrafast bulk photovoltaic response in van der Waals

heterostructures” presents a detailed investigation into the bulk photovoltaic effect (BPVE) in MoS₂/BP heterostructures, leveraging dual interfacial polarizations. This research stands out for its demonstration of significantly faster intrinsic BPVE response times compared to traditional non-centrosymmetric materials, offering a promising pathway for high-speed photodetection applications. The study’s novel approach to manipulating dual interfacial polarizations in vdW heterostructures, using MoS₂ and black phosphorus (BP), has led to the development of a device with an exceptionally high bulk photovoltaic coefficient. This advancement not only contributes to our understanding of BPVE in nanoscale systems but also opens up new possibilities for designing efficient photovoltaic devices. While the potential for publication in Nature Communications is high, I recommend addressing the following points for clarification:

1. The abstract suggests an “orders of magnitude faster” response compared to conventional materials, yet there are existing reports of picosecond responses in 2D materials. This comparison might be misleading and needs clarification.
2. The authors observe a slower response in the in-plane shift current, particularly when compared to the out-of-plane response. Could the authors offer a fundamental explanation for this observation? This paper might be related ([arXiv:2207.03772](https://arxiv.org/abs/2207.03772)).
3. A detailed physical picture of the ultrafast photocurrent process would be beneficial. Is the fast response attributed to ultrafast charge transfer due to type-II band alignment between BP and MoS₂ (ref. 27)? While the in-plane polarization in MoS₂/BP and WSe₂/BP heterostructures appears similar (Supplementary Section 8), the slow exponent observed in WSe₂/BP is absent in MoS₂/BP. Could there be an unconsidered slow time constant during fitting?
4. In the Supplementary Section 9, it’s noted that the photocurrent increases with the thickness of BP in the WSe₂/BP heterostructure. Could the authors elucidate the reasons behind this trend?
5. In Figures 2b and c, marking the overlap region of the heterostructure would enhance clarity.
6. Figure 3 should include information about the probing positions for the three devices.
7. For Figures 4d and e, varying the color scheme for data from different experimental conditions could improve contrast and readability.

Response to reviewer's comments

We greatly appreciate the reviewers' insightful comments which were very helpful for the improvement of our manuscript. In response to these, we provide point-by-point responses along with the modifications made in the revised manuscript.

Point-by-point responses to the issues raised by the reviewers:

Reviewer: 1

Bulk photovoltaic effect in vdW materials attracts great attention in recent years. In this work, authors built 1L-TMDCs/BP heterostructures, reproduced previous observation of BPVE in 1L-TMDCs/BP, and demonstrated ultrafast bulk photovoltaic response. Although, the BPVE in 1L-TMDCs/BP heterostructures has been demonstrated in previous works by Iwasa et al., authors provided some interesting results from a new perspective: the ultrafast response speed due to both existence of in-plane and out-plane polarization in the MoS₂/BP heterostructure. Major revisions should be made and data analysis should be strengthened before it can be considered for publications. Here is my comments.

Reply: We thank the reviewer for the positive remarks and constructive suggestions. We have carefully revised the manuscript by taking the suggestions and comments.

Question 1. Authors claim that intrinsic response time of WSe₂/BP structure is independent of the thickness of BP. However, from Fig. 3c, it is obvious that the response time also increases with increasing thickness of BP.

Reply: We thank the reviewer for pointing out this important issue. According to our DFT simulation (Fig. 2i), the generated in-plane polarization distributes in the TMD monolayer, which indicates that theoretically the BPVE is only contributed by the TMD monolayer and a few atomic layers of BP from the heterointerface. In Fig. 3c (which is now Fig. 3d in the revised manuscript), the experimentally obtained intrinsic response time of WSe₂/BP structure increases slightly with a relatively large error bar. However,

compared to the trend of the pure BP structure and MoS₂/BP structure, this increase in response time is much smaller. Regarding this phenomenon, we consider that due to the relatively large penetration depth of the excitation laser (780 nm), the charge carriers in the bottom of BP layers can be excited. Although these parts of the charge carrier have a negligible influence on the BPV current intensity, they could interact with the carriers from the interface and slightly affect its dynamics, which leads to a greater fluctuation in the photocurrent dynamic curves for WSe₂/BP structures than others. Thus, we consider this relatively small increase to be overall consistent with the hypothesis of a BPVE in WSe₂/BP structures, for which one would expect a thickness-independence.

In the revised manuscript, we have amended the corresponding discussions to avoid potential misunderstandings. The edited sentences are as follows.

“In contrast to the prominently extended response time in pure BP, the BPVE (Supplementary Section 10) dynamics from WSe₂/BP heterostructures demonstrated intrinsic response times of approximately 1 ns (upper panel in Fig. 3b and orange dots in Fig. 3d), which are not sensitive to the thickness of BP. This phenomenon can be understood as the generation of the photocurrent is due to a broken-symmetry-induced interfacial behavior since the in-plane polarization only distributes within a few atomic layers from the heterointerface¹⁵. After excitation, the charge carriers from the interface are separated by the in-plane polarization and form a BPVE. The slightly increased response time for over 40 nm WSe₂/BP heterostructures could be attributed to the influence of the increased amount of generated charge carriers in the bottom BP, which may affect the dynamics of carriers originating from the BPVE.”

Question 2. Did author measure the KPFM line scan profile of WSe₂/BP? It there any potential difference?

Reply: We thank the reviewer for the question. In response to this question, we have performed the requested KPFM and now provide the line scan as new data in

Supplementary Figure 4. According to our result, there is a potential difference between monolayer WSe₂ and BP. For a better comparison of the KPFM measurements, we fabricated 1L WSe₂/BP and 1L MoS₂/BP heterostructures in one sample (Supplementary Fig. 4a). The experimental results (Supplementary Fig. 4b) show that not only the potential difference between WSe₂ and BP is 30 mV which much smaller than that between MoS₂ and BP, but also the Fermi level of WSe₂ is lower than that of BP, which is consistent with our DFT simulation (Supplementary Fig.1c).

In the revised manuscript, we have revised the related sentence and have added a new Supplementary Section 4 to show the relative potential difference for 1L WSe₂/BP and 1L MoS₂/BP heterostructures. The added section and the revised sentence are as follows.

Supplementary Section 4: KPFM result and carrier extraction analysis for 1L WSe₂/BP heterostructure

To confirm our DFT calculation and show the uniqueness of the MoS₂/BP heterostructure, we fabricated 1L WSe₂/BP and 1L MoS₂/BP heterostructures in one sample (Supplementary Fig. 4a). In the KPFM line profile (Supplementary Fig. 4b), the potential difference between WSe₂ and BP is 30 mV which is much smaller than that between MoS₂ and BP and the Fermi level of WSe₂ is lower than that of BP. The result indicates that a p-p junction is formed in the WSe₂/BP heterostructure and the out-of-plane polarization points from BP to WSe₂, which is consistent with our DFT calculation (Supplementary Fig. 1c). In this case, though the formed out-of-plane polarization favors a hole transfer from BP to monolayer WSe₂, the type-I band alignment (Supplementary Fig. 4c) would suppress this effect. Hence, the out-of-plane polarization in the WSe₂/BP heterostructure would not accelerate the carrier extraction.

Supplementary Figure 4. KPFM result for 1L WSe₂/BP and 1L MoS₂/BP heterostructures. *a*, Optical image of 1L WSe₂/BP and 1L MoS₂/BP heterostructures in one sample. The dashed lines in red, green and dark yellow highlight the position of 1L WSe₂, BP and 1L MoS₂, respectively. *b*, KPFM line profile showing the relative potential of the three materials. The scanned line corresponds to the black line in shown *a*. *c*, Calculated energy band structures of the WSe₂/BP heterostructure.

“As a comparison, an armchair direction aligned WSe₂ monolayer/BP heterostructure device with the same configuration was also prepared, where the out-of-plane charge carrier extraction is unfavored (Supplementary Section 4).”

Question 3. *In abstract, authors claim that “the quantum tunneling assisted by the”. How does quantum tunneling phenomenon occur in the out-of-plane direction? I cannot see any quantum tunneling effect in this device configuration.*

Reply: We thank the reviewer for pointing out this. In the revised manuscript, we have given more detailed descriptions to avoid misunderstandings. The edited sentence is as follows.

“Here, we propose to realize ultrafast BPVE in vdW heterostructures by making use of dual interfacial polarizations, that is, the in-plane polarization gives rise to the BPVE in the overlayer and the charge carrier transfer assisted by the out-of-plane polarization further accelerates the interlayer electronic transport and enhances the BPVE.”

Question 4. For characterizing the response time of BP, MoS₂/BP, and WSe₂/BP devices, authors should provide clear characterization condition, such as bias voltage. If there is, will bias voltage affect the response time?

Reply: We thank the reviewer for this comment. All response time measurements in BP, MoS₂/BP, and WSe₂BP devices were conducted under zero external bias. For MoS₂/BP and WSe₂BP devices, the spontaneous photocurrent was collected at the center of the heterostructures, which originated from the BPVE. For two-terminal BP devices, the response time measurements were conducted at the junction between the electrode and the material, where a spontaneous photovoltaic effect was generated from the Schottky barrier at the junction.

In the revised manuscript, we have given a more detailed description of the experimental conditions. The edited sentences are as follows.

“The spontaneous photocurrent originated from BPVE in MoS₂/BP and WSe₂/BP devices was collected at the center of the heterostructures while the spontaneous photocurrent derived from Schottky barrier in pure BP two-terminal devices was collected at the edge of electrodes (red dots in Figure 3c). All intrinsic response time measurements were conducted at zero external bias.”

Question 5. What happens for response speed using above-bandgap lasers?

Reply: We thank the reviewer for the question. We expect that the response speed using above-MoS₂ monolayer bandgap laser is similar to the case for sub-MoS₂ monolayer bandgap illumination but a new slow photocurrent dynamic component may emerge.

First, we would like to discuss the intrinsic photocurrent dynamics for sub-MoS₂ monolayer bandgap illumination. Upon illumination of the 780 nm laser, the electrons generated from the BP layer can be transferred into the monolayer MoS₂ under the vertical built-in electric field. After that, the formed in-plane polarization further drives

these electrons into a BPV current. Due to the much faster response speed compared to that of the WSe₂/BP, we believe this process to be dominated by the ultrafast carrier transfer between MoS₂ and BP. Based on this, we propose a charge carrier dynamic formula to describe the intrinsic response: $\tau^{-1} = (\tau_d + \tau_s)^{-1} + \tau_r^{-1} + \tau_t^{-1}$, where τ represents the measured intrinsic response time, τ_d represents the charge carrier drift time, τ_r represents the charge carrier recombination time, τ_s represents the charge carrier transfer time from the semiconducting channel to the electrodes and τ_t^{-1} represents the charge carrier transfer between the MoS₂ and BP layers. In the sub-MoS₂ monolayer bandgap illumination, only one component of the photocurrent dynamics was observed, which could be due to the high efficiency of the carrier transfer.

On the other hand, for the above-MoS₂ monolayer bandgap illumination, the charge carrier transfer between MoS₂ and BP still exists. Thus, we argue that the above formula describing the intrinsic response time below MoS₂ bandgap illumination would also be suitable for the above-bandgap case. Due to the much faster speed of the charge carrier transfer than those of the drift and the recombination processes, the intrinsic response time would still be dominated by the charge carrier transfer and the tens of picoseconds response time would appear. In contrast, under above-bandgap illumination, because the charge carriers can be generated at the heterointerface, the in-plane polarization induced charge carrier drift process would occur simultaneously with the out-of-plane charge carrier transfer. In this case, another photocurrent dynamic component similar to the one in WSe₂/BP heterostructures may also emerge in MoS₂/BP.

Question 6. In method section, authors claim that the heterostructures were built after Raman characterization of BP flakes. Since surfaces of BP flakes are immediately oxidized when touching the air., how can authors guarantee a clear and oxidation-free interface which is very crucial for BPVE observation.

Reply: We agree with the reviewer that the BP flakes are relatively fragile and the

interface between BP and TMD is crucial for BPVE. There have been several experiment results demonstrate that there are oxidation-free interfaces in our heterostructures. For instance, a previous study [*J. Am. Chem. Soc.* 139, 10432–10440 (2017)] found that a thin layered BP would oxidate in 30 minutes at ambient conditions after exfoliation, and a Raman intensity ratio of A^1_g/A^2_g can be used to monitor the oxidation process. In detail, for the Raman spectrum collected with the laser polarization parallel to the zigzag direction of BP, a non-oxidated sample would have the largest Raman intensity ratio of $A^1_g/A^2_g \approx 0.97$. With ongoing oxidation time, the Raman intensity ratio of A^1_g/A^2_g would decay exponentially. For our sample fabrication, we did our heterostructure stacking immediately after the Raman characterization. Meanwhile, the calculated Raman intensity ratio of A^1_g/A^2_g under parallel zigzag direction configuration (Supplementary Fig. 2b, black) is a little bit higher than 0.97 (0.972). In addition, the BPVE was observed in every one of our devices, where it not only demonstrates a directional photocurrent response but also varies in intrinsic dynamics for different structures. Moreover, we note that for short-circuit current density versus power density results in different BPV devices (Supplementary Fig. 24), the performance of our WSe₂/BP is similar to the result reported by Ref. [*Science*, 372, 68-72 (2021)]. All of the above results indicate non-oxidated heterostructures and a highly efficient BPVE.

Question 7. Can author explain why the value of BPVE photocurrent is maximum in the middle region of the channel?

Reply: Yes, we can. The generation of the BPVE in our heterostructures relies on the formed in-plane polarization rather than the Schottky barrier, so the photocurrent intensity should be the same for the whole heterostructure region. Besides the contribution of in-plane polarization, the photocurrent intensity is also related to the charge carrier collection. In the middle region of the channel, the carrier collection length reaches a minimum value at the same time for the electrodes on both sides, which contributes to the highest carrier collection efficiency. Hence the photocurrent shows

the highest intensity in this region.

Question 8. How many layers of BP for DFT calculations? Will it affect the calculation results and conclusions?

Reply: We used a BP monolayer in the calculations of TMD/BP heterostructures. Our consideration is based on that the in-plane polarizations of the TMD monolayers are induced by symmetry breaking due to the interface. Whereas, the out-of-plane polarizations are determined by the difference in the work functions between them. Increasing the thickness of BP will not change them much. Therefore, in this sense, a monolayer is sufficient. In response to reviewer's question, we have performed calculations for the heterostructures with thicker BP films, i.e., TMD/BP-bilayer and TMD/BP-trilayer (now is Supplementary Section 18). Our results show that the trend in both the induced in-plane and out-of-plane polarizations remain unchanged (see the figure below). Our results on WSe₂/BP are also consistent with a previous study [see Fig. S8 in the Supplement Material of Ref. 15, Science 372, 68-72 (2021)].

In the revised manuscript, we have added the detailed BP layer information for calculations in the method and added a new Supplementary Section to discuss the influence of the BP layers on calculations and BPVE generation. The edited sentence and new section are as follows.

“A slab structure was used to model the BP monolayer, TMD monolayers, and their heterostructures.”

“Supplementary Section 18: Discussion of the effects of the thickness of BP layers on the polarizations

The calculations of the in-plane polarization, out-of-plane polarization, and charge carrier redistribution in the main text and above sections are based on the monolayer

BP. Though these calculations are consistent with our experimental results, most of our heterostructures were fabricated with thicker BP flakes. Here, we discuss the effects of the thickness of the BP layers on the polarizations. We consider two different situations. For the MoS₂/BP heterostructures, our calculations show that using a BP bilayer and trilayer only gives rise to slight changes in the potential and the charge carrier redistribution from the case using a BP monolayer. For instance, the magnitude of the in-plane polarization only changes from 0.025 eÅ for MoS₂/BP-1L to 0.023 eÅ for MoS₂/BP-2L. The out-of-plane polarization also remains almost unchanged. We obtain a similar trend for the MoS₂/BP-3L heterostructure. Based on these observations, we expect that using a BP monolayer is sufficient for demonstrating the physics of the induced polarizations in MoS₂/BP.

Besides, according to a previous report¹³, with the increase in BP thickness, its Fermi level would gradually move to the valance band and the BP layers become p-doped. Thus, the out-of-plane polarization direction also remains unchanged with the BP thickness. In summary, we are confident that our calculations based on monolayer BP would not affect the conclusions for BPVE generation in heterostructures with thicker BP.

Supplementary Figure 25. DFT calculations of MoS₂/BP heterostructures with 2L (a-b) and 3L (c-d) BP. a and c, Planar averaged electrostatic potential. b and d, Planar

averaged polar charge density.”

Edited Supplementary Reference:

13. Liu, X. C., et al. *Modulation of Quantum Tunneling via a Vertical Two-Dimensional Black Phosphorus and Molybdenum Disulfide p–n Junction. ACS Nano* **11**, 9143-9150 (2017).

*Question 9. Authors may include and discuss two recent works about BPVE in vdW materials and heterostructures Nat commun*14, 4230, 2023 *and Nat commun* 15, 501, 2024.

Reply: We thank the reviewer for suggestion and for providing the related references. We find that they are useful for our motivation. In the revised manuscript, we have added them in related discussion in the main text. The edited sentence and references are as follows.

“While due to the directional carrier transporting along nano size tube or the quantum well, one-dimensional vdW structures¹⁹⁻²¹ are reported to have high performance BPVE”

Edited References:

20. Zhou, Y., et al. *Giant intrinsic photovoltaic effect in one-dimensional van der Waals grain boundaries. Nat. Commun.* **15**, 501 (2024).

21. Liang, Z., et al. *Strong bulk photovoltaic effect in engineered edge-embedded van der Waals structures. Nat. Commun.* **14**, 4230 (2023).

Reviewer: 2

The paper “Dual polarization enabled ultrafast bulk photovoltaic response in van der Waals heterostructures” presents a detailed investigation into the bulk photovoltaic effect (BPVE) in MoS₂/BP heterostructures, leveraging dual interfacial polarizations. This research stands out for its demonstration of significantly faster intrinsic BPVE response times compared to traditional non-centrosymmetric materials, offering a promising pathway for high-speed photodetection applications. The study’s novel approach to manipulating dual interfacial polarizations in vdW heterostructures, using MoS₂ and black phosphorus (BP), has led to the development of a device with an exceptionally high bulk photovoltaic coefficient. This advancement not only contributes to our understanding of BPVE in nanoscale systems but also opens up new possibilities for designing efficient photovoltaic devices. While the potential for publication in Nature Communications is high, I recommend addressing the following points for clarification:

Reply: We thank the reviewer for the positive remarks and constructive suggestions. We have carefully revised the manuscript by taking the suggestions and comments.

***Question 1.** The abstract suggests an “orders of magnitude faster” response compared to conventional materials, yet there are existing reports of picosecond responses in 2D materials. This comparison might be misleading and needs clarification.*

Reply: We are sorry for the misleading. In the original manuscript, we considered that the intrinsic response time of our MoS₂/BP heterostructure was “orders of magnitude faster” than that of conventional bulk non-centrosymmetric crystals.

In the revised manuscript, we have modified the description to avoid misleading. The edited sentence is as follows.

“We illustrate the concept in MoS₂/black phosphorus heterostructures, where the

experimentally observed intrinsic BPVE response time achieves 26 ps, much faster than that of conventional bulk non-centrosymmetric materials.”

Question 2. The authors observe a slower response in the in-plane shift current, particularly when compared to the out-of-plane response. Could the authors offer a fundamental explanation for this observation? This paper might be related (arXiv:2207.03772).

Reply: We thank the reviewer for the question. We assume the reviewer is talking about our TRPC experiments (Figure 3d), which show that the in-plane shift current response in WSe₂/BP is roughly 50 times slower than that of the out-of-plane response in MoS₂/BP. Regarding this phenomenon, we consider that the out-of-plane response is dominated by the charge carrier transfer at the heterointerface. Whereas, the in-plane shift current response is dominated by the in-plane charge carrier drift process. According to a previous investigation in *Adv. Funct. Mater.* 32, 2206952 (2022), the out-of-plane charge carrier transfer between thin layered BP and MoS₂ is at tens of femtosecond time scale. Due to the reduced charge carrier transfer efficiency for thicker layers, we would expect that the charge carrier transfer time extends to several picoseconds for heterostructures in our case.

On the other hand, the typical lateral channel length is 7 μm in our device configuration (Figure 2a), which is much larger than the vertical heterostructure thickness at tens of nanometers. In this case, the in-plane charge carrier drift process could be more influenced by the electron-electron and electron-impurity/defect scatterings, which can slow the in-plane shift current response as discussed in the recent paper (arXiv:2207.03772). It should be noted that, our intrinsic shift current response is based on measurements in conventional circuits, in which charge carrier dynamics are involved. This induces the response time in our case at picoseconds to a nanosecond time scale compared with the calculated shift current response at the femtoseconds time scale.

In the revised manuscript, we have added the reference and given a discussion in

the main text. The edited sentence and reference are as follows.

“The generation of shift current by the BPVE in non-centrosymmetric bulk materials and TMD has been investigated by the DFT calculations²⁹ and THz emission spectroscopy³⁰⁻³².”

Edited References:

29. He, F., Chen, D., Ren, X., Meng, S. & He, L. Ultrafast shift current dynamics in WS₂ monolayer. *Phys. Rev. Res.* **6**, 013123 (2024).

Question 3. A detailed physical picture of the ultrafast photocurrent process would be beneficial. Is the fast response attributed to ultrafast charge transfer due to type-II band alignment between BP and MoS₂ (ref. 27)? While the in-plane polarization in MoS₂/BP and WSe₂/BP heterostructures appears similar (Supplementary Section 8), the slow exponent observed in WSe₂/BP is absent in MoS₂/BP. Could there be an unconsidered slow time constant during fitting?

Reply: We thank the reviewer for the suggestion and questions. Regarding the ultrafast photocurrent process, we propose the detailed physical picture is as follows. For WSe₂/BP heterostructure without charge carrier transfer, because its in-plane polarization merely distributes within atomic layers from the heterointerface, the photo-generated charge carriers in these layers will be separated under the effect of in-plane polarization to form a BPV current. It should be noted that this BPVE can only occur at the above WSe₂ bandgap. In contrast, for MoS₂/BP heterostructure, though the monolayer MoS₂ cannot absorb the photon energy from the 780 nm laser, the photo-generated charge carriers in BP can be transferred into the monolayer MoS₂ with the help of the out-of-plane built-in field. In this case, the formed in-plane polarization further drives these electrons into a BPV current. In addition, the remaining holes in the bottom BP can be collected by the electrodes as the electrodes contact the BP flake in our device configuration, thus that the carrier circulation can be realized.

Regarding the picosecond photocurrent generation dynamics in the MoS₂/BP heterostructures, we attribute them to the ultrafast charge transfer between MoS₂ and

BP due to both of the type-II band alignment and out-of-plane built-in field. The intrinsic photocurrent dynamics in MoS₂/BP heterostructures demonstrate single exponential decay, and we believe that there is no unconsidered slow time component during the fitting. We propose to use the formula: $\tau^{-1} = (\tau_d + \tau_s)^{-1} + \tau_r^{-1} + \tau_t^{-1}$, to describe the intrinsic photocurrent dynamics in MoS₂/BP heterostructures. Here, τ represents the measured intrinsic response time, τ_d represents the charge carrier drift time, τ_r represents the charge carrier recombination time, τ_s represents the charge carrier transfer time from the semiconducting channel to electrodes and τ_t^{-1} represents the charge carrier transfer between MoS₂ and BP layers. According to this formula, because the charge carrier transfer time between MoS₂ and BP is much faster than the charge carrier drift time and recombination time, it dominates the intrinsic photocurrent response and the photocurrent dynamics demonstrates a single exponential decay. Similar photocurrent dynamic phenomena were also observed in graphene/WSe₂/graphene heterostructures, according to previous publications in *Nat. Nanotechnol.* 11, 42-46 (2016) and *Nat. Commun.* 7, 12174 (2016). Under the above-WSe₂-bandgap illumination, the graphene/28nm-WSe₂/graphene heterostructures demonstrate an intrinsic response time of approximately hundreds of picoseconds, reflecting the charge carrier drift time in WSe₂ (*Nat. Nanotechnol.* 11, 42-46 (2016)). In contrast, under sub-WSe₂-bandgap illumination, photocurrent dynamics for the same heterostructure demonstrates a single exponential decay and the intrinsic response time is down to 1.3 picoseconds, reflecting the hot carrier injection from graphene to WSe₂ (*Nat. Commun.* 7, 12174 (2016)).

In the revised manuscript, we have added a more detailed physical picture of photocurrent analysis at the related positions. The added sentences are as follows.

“This phenomenon can be understood as the generation of the photocurrent is due to a broken-symmetry-induced interfacial behavior since the in-plane polarization only distributes within a few atomic layers from the heterointerface.¹⁵ After excitation, the charge carriers from the interface are separated by the in-plane polarization and form a BPVE.”

“With the type-II band alignment and vertical p-n junction in the heterostructures, the electrons generated from the BP layer by the illumination are rapidly transferred into the monolayer MoS₂ under the out-of-plane polarization. After that, the formed in-plane polarization can further accelerate these electrons via the BPVE. Because the electrodes parallel to the mirror plane also contact the bottom BP flake in our structures, the remaining holes in the bottom BP are collected by the electrodes and the carrier circulation is realized. In this way, the BPVE response for MoS₂/BP heterostructure is related to the BP thickness, and its ultrafast speed is attributed to the charge carrier transfer.”

Question 4. In the Supplementary Section 9, it's noted that the photocurrent increases with the thickness of BP in the WSe₂/BP heterostructure. Could the authors elucidate the reasons behind this trend?

Reply: We thank the reviewer for the comment. For the WSe₂/BP heterostructures without charge carrier transfer, the in-plane polarization distributes at the monolayer WSe₂, and the BPVE is only related to atomic layers from the heterointerface. Hence, theoretically the generated photocurrent intensity should be independent of the BP thickness. In our WSe₂/BP heterostructures with the BP thickness of 10 nm, 17 nm, and 40 nm, the BPV current intensity from the SPCM is 800 pA 1.5 nA and 1.6 nA, respectively. According to previous results in *Science*, 372, 68-72 (2021), the photocurrent intensity for the 68 nm sample is 1.5 times higher than that of 19 nm. Hence, we think our results are comparable, and especially for the sample with 17 nm and 40 nm BP, the BPV current intensity is almost the same. Regarding the slightly increased photocurrent intensity for thicker heterostructure, one possible reason is that the photovoltaic effect from the bottom BP also contributes to the overall current collection. While, for the 4 nm sample, because we conducted our measurements in an ambient condition, we used a much lower excitation power for the SPCM measurements to avoid its degeneration.

In the revised supplementary information, we have clarified the different experimental conditions to avoid a misunderstanding. The edited sentence is as follows.

“To prevent degradation of heterostructures with thin BP, a lower excitation power was used for the sample with 4 nm thick BP.”

Question 5. In Figures 2b and c, marking the overlap region of the heterostructure would enhance clarity.

Reply: We thank the reviewer for the suggestion. In the revised manuscript, we have marked the overlap region of the heterostructure in Figures 2b and 2c by the gray dashed lines and noted it at the corresponding Figure caption to enhance the clarity. The edited figure and figure caption are as follows.

b and c, SPCM images with the electrodes E1-E2 (b) and E3-E4 (c) with the 780 nm laser. The dashed gray lines highlight the overlap region of the heterostructure. The black scale bars are 4 μm .

Question 6. Figure 3 should include information about the probing positions for the three devices.

Reply: We thank the reviewer for the suggestion. In the revised manuscript, we have added the images and highlighted the probing positions in three representative devices in Figures 3 and noted them at the corresponding Figure caption to enhance the clarity. The edited figure and figure caption are as follow.

c, Representative optical images in different structures to show the experimental conditions for TRPC measurements with electrodes E1 and E2. The electrodes E1 and E2 are parallel to the mirror plane and armchair direction for heterostructures and BP, respectively. The red dots indicate the position for conducting the TRPC measurement. All measurements were conducted at zero external bias.

Question 7. For Figures 4d and e, varying the color scheme for data from different experimental conditions could improve contrast and readability.

Reply: We thank the reviewer for this suggestion. In the revised manuscript, we have changed the color scheme for data from different experimental conditions to improve contrast and readability. The edited figure is as follows.

Reviewer: 3

This paper reports on the bulk photovoltaic effect (BPVE) in integrated heterostructures of two van der Waals compounds with different crystalline symmetry. In particular, MoS₂/BP heterostructures, which possess both in-plane spontaneous polarization and out-of-plane built-in field, exhibit BPVE with sizable magnitude and ultrafast response time. The result may have some importance in the device application. However, the emergence of in-plane polarization and BPVE at van der Waals heterointerfaces have already been reported (Ref. 15). Furthermore, although the authors emphasize the importance of the out-of-plane built-in field, there is no clear and reasonable explanation for why it enhances the magnitude and speed of in-plane photocurrent response. Therefore, I cannot recommend the publication of the present form of the manuscript in Nature Communications. Other points of concern are noted below.

Reply: We appreciate the reviewer rates that our results may have some importance in the device application. During this revision, we have carefully followed the reviewer's comments and suggestions in the revised manuscript.

Concerning the novelty of our work, this work is distinct from the van der Waals heterointerfaces work reported in Reference 15. Our work has introduced additional out-of-plane polarization that not only provides a new device configuration, but also leads to efficient manipulating of photogenerated carriers, realizing ultrafast response speed and an exceptionally high bulk photovoltaic coefficient which are beyond the previous work with pure in-plane polarization. We believe that our work represents an important advancement from a new perspective: the ultrafast response speed due to the dual polarization along with the high BPV efficiency.

Regarding the enhanced response speed and photocurrent magnitude in MoS₂/BP structure, here we give a detailed explanation. Firstly, we want to point out that our device configuration is different from the one for conventional p-n junction. For our MoS₂/BP heterostructure devices, all electrodes were fabricated on top of the monolayer MoS₂ rather than on respective p-type and n-type materials in conventional

p-n junction devices. Hence, the difference between the cathode and the anode in our device configuration is not related to the built-in field for the p-n junction but to the direction of the in-plane polarization. In this case, the out-of-plane polarization is not responsible for enhancing the response speed of in-plane photocurrent, and the picosecond intrinsic response time is attributed to the ultrafast charge carrier transfer between BP and MoS₂. In detail, with the type-II band alignment and vertical p-n junction in the heterostructures, the electrons generated from the BP layer by illumination with 780 nm can be rapidly transferred into the monolayer MoS₂ under the out-of-plane polarization. After that, the formed in-plane polarization can further drive these electrons into a BPVE. It should be noted that, the BPVE still originates from the in-plane polarization, and out-of-plane polarization only serves to extract more electrons from BP to the MoS₂. Compared to the WSe₂/BP heterostructure that only takes advantage of the charge carriers at the interface, the increased number of electrons from the bottom layer BP hence enhances the BPVE in MoS₂/BP.

In the following, we have provided point-by-point responses with more detailed analyses for the comments raised by the reviewer and have added modifications in the revised manuscript.

Question 1. BPV effect, or shift current generation, does not require the presence of spontaneous polarization. Materials belonging to any noncentrosymmetric space groups should have finite second-order susceptibility tensor components. The magnitude of the BPE response is also irrelevant to the presence of spontaneous polarization. The present manuscript does not adequately address why symmetry reduction is necessary by creating integrated heterostructures to induce/enhance BPE.

Reply: We thank the reviewer for the comment. We agree with the reviewer that the BPVE does not require the presence of spontaneous polarization and that the magnitude of the BPE response is also irrelevant to the presence of spontaneous polarization. Previously, we meant the inversion symmetry breaking for the generation of BPVE,

which led us to present symmetry analysis for TMD monolayers in Fig. 1. In the presence of layers that has a different symmetry from the TMD monolayers in heterostructures, symmetry reduction is expected. The D_{3h} symmetry can be reduced to two nonpolar groups and five polar groups in heterostructures. While systems with polar groups have spontaneous polarizations, which are good for the generation of BPVE.

In the revised manuscript, to avoid misleadings, we have modified the related sentence. The edited sentence is as follows.

“The generation of BPVE relies on the broken symmetry in non-centrosymmetric materials, for which polar groups can be introduced.”

Regarding the second comment, we have three main reasons for constructing heterostructures to realize enhanced BPVE, which are the flexibility in material selection, the improvement of BPVE performance, and the versatility in device design.

(1) Flexibility in material selection. Compared with finding the symmetry-naturally-broken low-dimensional materials, heterostructure construction provides an easier method for the combination of various materials with complementary properties to reduce the symmetry and realize BPVE. In our MoS₂/BP heterostructure, both of the constituents are centrosymmetric. However, according to group theory, the combination of these two centrosymmetric materials can easily break the rotation symmetry and form an in-plane polarization to generate BPVE.

(2) Improvement of BPVE performance. By choosing materials with suitable band alignment and conductive type, the heterostructures can be engineered to have tailored structures that efficiently reduce the charge carrier recombination and enhance the photocurrent. In our MoS₂/BP heterostructure, the combination of n-type MoS₂ and p-type BP realizes an out-of-plane polarization, which aids the generated charge carriers in their transfer from BP to the monolayer MoS₂ and affords a highly efficient BPVE performance in our heterostructure (Figure. 5). In addition, this charge carrier transfer

behavior also broadens the photoresponse of the heterostructure to the near-infrared region.

(3) Versatility in device design. The BPVE can be implemented in various device applications, such as solar cells and photodetectors. The heterostructure construction provides the possibility of designing BPVE devices for specific application requirements. For our MoS₂/BP heterostructure, its fast intrinsic and extrinsic response time enables a highly efficient photodetection application. Meanwhile, it may have an advantage over other 2D BPV devices in a relatively large photosensitive area.

In the revised manuscript, we have added the above discussion in the main text. The edited sentences are as follows.

“Besides, combining the different vdW materials with complementary properties, such as conductive types, to form heterostructures and engineering the low-symmetric interfaces that facilitate the charge extraction may enable a further progress of the BPVE and nano devices for a specific application.”

“Besides strain engineering²² and effective dimensionality reduction²³, such broken symmetries can be easily achieved by building heterostructures.”

Edit References:

22. Liang, J., et al. Monitoring Local Strain Vector in Atomic-Layered MoSe₂ by Second-Harmonic Generation. *Nano Lett.* **17**, 7539-7543 (2017).
23. Cysne, T. P., Guimarães, F. S. M., Canonico, L. M., Costa, M., Rappoport, T. G. & Muniz, R. B. Orbital magnetoelectric effect in nanoribbons of transition metal dichalcogenides. *Phys. Rev. B* **107**, 115402 (2023).

Question 2. The photoresponse speed of WSe₂/BP device does not depend on the thickness of BP, whereas it depends on the thickness in pure BP and MoS₂/BP devices. These results are confusing for me. First, why does the pure BP exhibit spontaneous photocurrent? It will be in centrosymmetric crystalline structure. Second, why do the

WSe₂/BP and MoS₂/BP devices show different thickness dependencies? The authors interpret that photocurrent is generated only at the heterointerface in the WSe₂/BP device, whereas photocurrent is generated deep inside of BP in the MoS₂/BP devices and transferred by the perpendicular built-in field. However, the authors consider that the origin of the photocurrent in MoS₂/BP device is shift current driven by the interfacial in-plane polarization (in line 98). These conclusions seem to be completely conflicting. The interface-induced inversion symmetry breaking will occur only a few atomic layers from the interface.

Reply: We are sorry for the caused confusion. Below are our explanations.

For pure BP two-terminal devices, we conducted the photoresponse measurements at the edge of the electrode, where a Schottky barrier between BP and the metal gold leads to a spontaneous photovoltaic (PV) effect. This effect is common for photocurrent investigations in microscopic area, for example, *Nat. Commun.* 6, 8831 (2015) and it does not require a broken symmetry.

In the revised manuscript, we have given a more detailed description of the photocurrent measurement conditions. Meanwhile, we have added the image which highlights the probing position for the BP two-terminal device in Figure 3 and noted it in the corresponding Figure caption to enhance the clarity. The edited sentences, figure, and figure caption are as follows.

“The spontaneous photocurrent derived from BPVE in MoS₂/BP and WSe₂/BP devices was collected at the center of the heterostructures while the spontaneous photocurrent derived from Schottky barrier in pure BP two-terminal devices was collected at the edge of electrodes (red dot in Figure 3c).”

Figure 3c, Representative optical images in BP showing the signal collection position at red dot.

Concerning the difference in the thickness-dependence of the photoresponse for WSe₂/BP and MoS₂/BP, we can explain a bit more. Firstly, we attribute both of the photocurrent generation mechanisms in WSe₂/BP and MoS₂/BP heterostructures to the BPVE which occurs at the heterointerface. For our device configuration, all the electrodes were fabricated on top of a TMD monolayer, and in this case, the in-plane-polarization-induced BPVE was experimentally demonstrated by the scanning photocurrent microscope results (Figure. 2b-c and Supplementary Section 10). Towards the different thickness dependence of intrinsic response time, we interpret it as the photocurrent generation dynamic difference related to the out-of-plane built-in field.

For WSe₂/BP without a charge carrier transfer, because the in-plane polarization distributes at the atomic layer from the heterointerface (Figure 2i), the BPVE is a kind of interfacial behavior and its dynamics display an insensitive BP-thickness dependence.

In contrast, with the out-of-plane built-in field and type-II band alignment in MoS₂/BP heterostructures, there is an electric force to push the generated charge carriers deep inside the BP toward the MoS₂ monolayer and transfer to it. Hence, besides the charge carrier generated at the interface, the charge carriers generated deep in the BP flake also contribute to the photocurrent. However, it should be noted that the BPVE still originates from the in-plane polarization, which is verified by the anisotropic photocurrent response patterns (Figure. 2b-c). The out-of-plane polarization only serves to extract more electrons from BP to the MoS₂. For heterostructures with thicker BP

flake, the number of transferred electrons increases, thus the intrinsic response time extends from 26 ps to 60 ps (Figure 3b bottom panel and Figure 3d blue dots). Overall, based on our device configuration, the scanning photocurrent microscope results, the in-plane polarization distribution simulation, and charge carrier dynamics analyses, we consider BPVE is generated at the interface for both of the heterostructures but the out-of-plane built-in field leads to a BP-thickness dependence of intrinsic response time in MoS₂/BP.

Question 3. The enhanced photoresponse speed in MoS₂/BP heterostructures is attributed to the rapid transfer of photocarriers due to the built-in electric field perpendicular to the heterointerface. However, this interpretation seems unreasonable. For the cathode electrode, the built-in field pointing from BP to MoS₂ is effective in correcting the holes yielded deep inside of BP. Inversely, the built-in field will prevent the correction of electrons at the anode electrode. In total, it is unclear whether such a field has a positive effect on the carrier correction.

Reply: We thank the reviewer for this comment. We would like to clarify that our device configuration is different from the one for conventional p-n junction. This difference may lead to the readers misunderstand our results. In detail, for our MoS₂/BP heterostructure devices, both of the electrodes parallel to the mirror plane were fabricated on top of the monolayer MoS₂. This means that compared to the conventional p-n junction where the electrodes are on respective p-type and n-type materials, the difference between the cathode and the anode in our devices is not related to the built-in field for the p-n junction but the direction of the in-plane polarization. With the increase in BP thickness, due to the effect of the out-of-plane built-in field pointing from MoS₂ to BP, the number of electrons transferred from BP to monolayer MoS₂ increases. While, for the thicker samples, because the in-plane polarization is only related to the atomic layers from the heterointerface, its strength and direction stay almost unchanged. In this case, under the sub-MoS₂-monolayer bandgap illumination at zero-external bias, the out-of-plane built-in field extracts the generated electrons

from the bottom BP to MoS₂ monolayer and the in-plane polarization drives these extra charge carriers into BPV current. There is always a positive effect on electron extraction, and the enhancement of the BPVE is attributed to the increased number of electrons. For the remaining holes in the bottom layer BP, because the electrodes in our structures also contact the BP flake, they can also be collected by the electrodes, and hence the charge carrier circulation can be realized.

In the revised manuscript, we have modified the inset of Figure 1g and Figure 4a as well as the related sentences for a clearer illustration of our device configuration and analysis of the BPVE dynamics, respectively. The modified figures and sentences are as follows.

Figure 1g insert, The insert shows the schematic illustration of the TMD/BP BPV device.

Figure 4a, Schematic illustration of the TRPC measurement.

“With the type-II band alignment and vertical p-n junction in the heterostructures,

the electrons generated from the BP layer by the illumination are rapidly transferred into the monolayer MoS₂ under the out-of-plane polarization. After that, the formed in-plane polarization can further accelerate these electrons via the BPVE. Because the electrodes parallel to the mirror plane also contact the bottom BP flake in our structures, the remaining holes in the bottom BP are collected by the electrodes and the carrier circulation is realized. In this way, the BPVE response for MoS₂/BP heterostructure is related to the BP thickness, and its ultrafast speed is attributed to the charge carrier transfer.”

Question 4. The response time of shift current for ultrashort pulsed light excitation has been evaluated by THz emission spectroscopies (N. Laman et al., J. Appl. Phys. 98, 103507 (2005), L. Braun et al., Nat. Commun. 7, 13259 (2016), M. Sotome, et al., PNAS 116, 1929 (2019)). These works have demonstrated that the typical response time of shift current is about 0.1 ps, which is much faster than the intrinsic photocurrent response time of 26 ps obtained in this study. It will be necessary to mention the difference in the response speed and its possible reasons.

Reply: Regarding the difference in the shift current response speed, we consider the measurement mechanism and the measured signal are different between our TRPC technique and the THz emission spectroscopy. Our TRPC measurement technique is based on the recovery of a saturated photocurrent in a pump-probe configuration. The photocurrent response measured by this technique though excluding the extrinsic influences in the circuit, such as parasitic capacitance and RC-time, is still related to the charge carrier transport dynamics, for example, charge carrier recombination, charge carrier drift, and charge carrier transfer.

On the other hand, THz emission spectroscopy provides a transient photocurrent density, based on the propagation of an electromagnetic wave with a terahertz-frequency range. This technique does not require conventional electric circuits with electrodes, it neglects some of the charge carrier transport dynamics, for example,

charge carrier transfer from the materials to the electrodes, and hence demonstrates much faster results than that from our technique. It should be noted that, compared with the THz emission spectroscopy technique, because our TRPC technique can reflect an ultrafast upper limit of photocurrent dynamics in conventional electric circuits, we believe it holds the advantage in the understanding and guiding the design of micro-nano devices for practical application.

In the revised manuscript, we have added the above discussion into the main text. The added sentences are as follows.

“The generation of shift current by the BPVE in non-centrosymmetric bulk materials and TMD has been investigated by the DFT calculations²⁹ and THz emission spectroscopy³⁰⁻³². In these studies, the typical response time of the shift current was roughly 0.1 ps, much faster than our results. It should be noted that our TRPC technique is based on the photocurrent dynamic investigation in conventional electric circuits, where the overall charge carrier transport behavior depends on charge carrier recombination, charge carrier drift, and charge carrier transfer.”

Edited References:

30. Laman, N., Bieler, M. & van Driel, H. M. Ultrafast shift and injection currents observed in wurtzite semiconductors via emitted terahertz radiation. *J. Appl. Phys.* **98**, 103507 (2005).
31. Braun, L., et al. Ultrafast photocurrents at the surface of the three-dimensional topological insulator Bi₂Se₃. *Nat. Commun.* **7**, 13259 (2016).
32. Sotome, M., et al. Spectral dynamics of shift current in ferroelectric semiconductor SbSI. *Nat. Commun.* **11**, 1929-1933 (2019).

REVIEWER COMMENTS

Reviewer #2 (Remarks to the Author):

The authors have satisfactorily addressed all my questions. They have conducted experiments according to the suggestions from referee #1 and integrated the results into the revised manuscript. Furthermore, they have also provided detailed discussions in the revised manuscript for enhanced clarity. Therefore, I recommend this work for publication in Nature Communications.

Reviewer #3 (Remarks to the Author):

The reviewer thanks the authors for their courteous replies to all the comments and questions. However, the reviewer remains unconvinced as to the mechanism by which the out-of-plane polarization enhances BPVE in the MoS₂/BP heterostructure. According to the authors' explanation, the observed BPVE originates from the in-plane polarization at the heterointerface. The built-in out-of-plane electric field serves to extract more electrons from deep inside of BP. After that, the in-plane polarization further drives these electrons into BPVE. The reviewer's question is what the authors envision as the mechanism by which photocarriers corrected at the heterointerfaces are accelerated in one direction by in-plane polarization. In the shift current mechanism, displacement of electrons occurs during the optical transition process, and it does not have any contribution to the rectification of photocarriers remaining after the optical transition.

Point-by-point responses to the issues raised by the reviewers:

Reviewer: 2:

The authors have satisfactorily addressed all my questions. They have conducted experiments according to the suggestions from referee #1 and integrated the results into the revised manuscript. Furthermore, they have also provided detailed discussions in the revised manuscript for enhanced clarity. Therefore, I recommend this work for publication in Nature Communications.

Reply: We thank the reviewer for the careful review and constructive suggestion of our paper. We are glad that our revised manuscript has satisfied the reviewer and appreciate his/her positive comments and recommendation for publication of this work in *Nature Communications*.

Reviewer: 3:

The reviewer thanks the authors for their courteous replies to all the comments and questions. However, the reviewer remains unconvinced as to the mechanism by which the out-of-plane polarization enhances BPVE in the MoS₂/BP heterostructure. According to the authors' explanation, the observed BPVE originates from the in-plane polarization at the heterointerface. The built-in out-of-plane electric field serves to extract more electrons from deep inside of BP. After that, the in-plane polarization further drives these electrons into BPVE. The reviewer's question is what the authors envision as the mechanism by which photocarriers corrected at the heterointerfaces are accelerated in one direction by in-plane polarization. In the shift current mechanism, displacement of electrons occurs during the optical transition process, and it does not have any contribution to the rectification of photocarriers remaining after the optical transition.

Reply: We would like to thank the reviewer for his/her thoughtful comments and suggestions. We truly appreciate the time and effort that the reviewer invested in our paper. We carefully consider the reviewer's suggestion by making more clear clarification for the mechanism and the enhancement of the BPVE in the MoS₂/BP heterostructure. Below we would like to address the reviewer's concern in three aspects.

First, we want to explain a little bit about the details of our experiment. In our experiments on probe station, the observed spontaneous photocurrent induced by the un-focused laser (Figure 1g) in MoS₂/BP heterostructure was approximately 240 nA, which doubled that in WSe₂/BP heterostructure at approximately 120 nA. Besides, for photocurrent measurement in the micro-area, a spontaneous photocurrent appeared along the mirror plane for MoS₂/BP heterostructure under sub-TMD monolayer illumination. The above two phenomena confirm the effect of our newly introduced out-of-plane polarization in MoS₂/BP heterostructure for BPVE enhancement,

considering that the calculated in-plane polarizations are comparable for both of the heterostructure (Supplementary Sectional 9).

Second, there may be an additional mechanism to the BPVE in our heterostructure system. we agree with the reviewer in that in the shift current mechanism, displacement of electrons occurs during the optical transition process, and it does not have any contribution to the rectification of photocarriers remaining after the optical transition. Besides electron shift in real space, the BPVE is also related to the unbalanced velocity distribution in momentum space. One possible explanation is that the spontaneous photocurrent generation mechanism in our MoS₂/BP may originate from ballistic current. Under linearly polarized light, the two-band contributed ballistic current can be expressed by:

$$j^{q,\text{diag}} = \frac{\pi\tau_0 e^3}{\omega^2 \hbar} \text{Re} \left[\sum_{l,n} \sum_{\Omega=\pm\omega} \int_{BZ} \frac{d\mathbf{k}}{(2\pi)^3} (f_{l\mathbf{k}} - f_{n\mathbf{k}}) \right. \\ \left. \times v_{nl}^r(\mathbf{k}) v_{ln}^s(\mathbf{k}) v_{nn}^q(\mathbf{k}) \delta(\varepsilon_{n\mathbf{k}} - \varepsilon_{l\mathbf{k}} - \hbar\Omega) \right] E_r E_s \quad (1)$$

Normally, it only occurs in magnetic materials. This is because, for band structure with time-reversal symmetry, the three-velocity term $v_{nl}^r v_{ln}^s v_{nn}^q$ undergoes a sign reversal for $-\mathbf{k}$: $v_{nl}^r(-\mathbf{k}) v_{ln}^s(-\mathbf{k}) v_{nn}^q(-\mathbf{k}) = -v_{nl}^r(\mathbf{k}) v_{ln}^s(\mathbf{k}) v_{nn}^q(\mathbf{k})$, where the Fermi–Dirac function and the delta function in Eq. (1) will be even for \mathbf{k} and $-\mathbf{k}$, and the integration of \mathbf{k} over the Brillouin zone in will be zero. However, in non-magnetic materials with additional scattering processes, the above Eq. (1) can be rewritten into a form:

$$j^{q,\text{diag}} = 2e\tau_0 \sum_{cv\mathbf{k}} \Gamma_{cv,\mathbf{k}}^{rs}(\omega) [v_{c\mathbf{k}}^q - v_{v\mathbf{k}}^q], \quad (2)$$

where $\Gamma_{cv,-\mathbf{k}}^{rs}(\omega) \neq \Gamma_{cv,\mathbf{k}}^{rs}(\omega)$ if additional interaction is considered for non-centrosymmetric systems. In this case, the BPVE can occur during the flow and transfer of charge carriers after the optical excitation. For our MoS₂/BP heterostructure, the BPVE is not only contributed by the photo-excited charge carriers at the hetero-

interface but also related to the charge carriers transferred from the bottom BP. Hence, the charge carrier transfer process may introduce electron-defect, electron-phonon, or electron-electron interactions and contribute ballistic current. Besides, the indirect bandgap structure in the MoS₂/BP heterostructure may lead to additional scattering processes under sub-monolayer MoS₂ bandgap excitation at the same time.

Moreover, regarding the enhancement of BPVE in our MoS₂/BP, we attribute it to the increased charge carrier due to the transfer from BP layers for sub-monolayer MoS₂ bandgap excitation. For the above--monolayer MoS₂ bandgap excitation, we attribute the enhanced BPVE to the simultaneous generation of shift current by the charge carrier at the heterointerface and ballistic current by the charge carrier transfer.

We also want to express that the BPVE in 2D van der Waals heterostructures is relatively new and still under investigation. In contrast to the conventional 3D bulk ferroelectric materials with a single component, layered 2D materials and especially their heterostructures may introduce some new phenomena.

In the revised manuscript, we have revised the related sentence and included the above discussion in the main text. The edited sentences are as follows.

“With the increase in the input laser power, both devices display a linear to sub-linear (0.5) photocurrent transition at an excitation power of approximately 50 μW (Fig. 1h), which is different from the photovoltaic (PV) effect trend originating from the p-n junction or Schottky barrier.”

“Regarding the mechanism of spontaneous photocurrent, shift current is one of the main originations for BPVE. The generation of shift current in bulk non-

centrosymmetric materials and TMD has been investigated by the calculation²⁹ and THz emission spectroscopy³⁰⁻³², where its typical response time was roughly 0.1 ps. However, in the shift current mechanism, displacement of electrons occurs during the optical transition process, and it does not have a contribution to the BPVE after the optical transition. Besides electron shift in real space, the BPVE is also related to the unbalanced velocity distribution in momentum space, which is referred to ballistic current. Normally, under linearly polarized light excitation, the ballistic current can occur in non-centrosymmetric non-magnetic materials, if an additional scattering process is considered³³. For our MoS₂/BP heterostructures, the BPVE is not only contributed by the photo-excited charge carriers at the hetero-interface but also related to the charge carriers transferred from the bottom BP, where the charge carrier transfer process may introduce electron-phonon or electron-defect interactions and contribute ballistic current.”

Edited Reference:

[33] Dai, Z. & Rappe, A. M. Recent progress in the theory of bulk photovoltaic effect. Chem. Phys. Rev. 4, 011303 (2023).

REVIEWERS' COMMENTS

Reviewer #3 (Remarks to the Author):

The authors have adequately responded to the reviewer's question. The ballistic current mechanism reasonably explains the enhanced BPVE at the interface. Thus, I recommend publishing this paper in Nature Communications in the present form.